# Changing the Paradigm for Tractography Segmentation in Neurosurgery: Validation of a Streamline-Based Approach

**DOI:** 10.3390/brainsci14121232

**Published:** 2024-12-07

**Authors:** Silvio Sarubbo, Laura Vavassori, Luca Zigiotto, Francesco Corsini, Luciano Annicchiarico, Umberto Rozzanigo, Paolo Avesani

**Affiliations:** 1Department of Neurosurgery, “S. Chiara” University-Hospital, Azienda Provinciale per i Servizi Sanitari, 39122 Trento, Italy; 2Center for Mind/Brain Sciences (CIMeC), University of Trento, Via delle Regole, 101, Mattarello, 38123 Trento, Italy; 3Centre for Medical Sciences (CISMED), University of Trento, 38122 Trento, Italy; 4Department of Cellular, Computation and Integrative Biology (CIBIO), University of Trento, 38123 Trento, Italy; 5Department of Psychology, “S. Chiara” University-Hospital, Azienda Provinciale per i Servizi Sanitari, 39122 Trento, Italy; 6Department of Radiology, “S. Chiara” University-Hospital, Azienda Provinciale per i Servizi Sanitari, 39122 Trento, Italy; 7Neuroinformatics Laboratory (NiLab), Bruno Kessler Foundation (FBK), 39123 Trento, Italy

**Keywords:** brain mapping, bundle segmentation, glioma, presurgical planning, tractography, white matter

## Abstract

In glioma surgery, maximizing the extent of resection while preserving cognitive functions requires an understanding of the unique architecture of the white matter (WM) pathways of the single patient and of their spatial relationship with the tumor. Tractography enables the reconstruction of WM pathways, and bundle segmentation allows the identification of critical connections for functional preservation. This study evaluates the effectiveness of a streamline-based approach for bundle segmentation on a clinical dataset as compared to the traditional ROI-based approach. We performed bundle segmentation of the arcuate fasciculus, of its indirect anterior and posterior segments, and of the inferior fronto-occipital fasciculus in the healthy hemisphere of 25 high-grade glioma patients using both ROI- and streamline-based approaches. ROI-based segmentation involved manually delineating ROIs on MR anatomical images in Trackvis (V0.6.2.1). Streamline-based segmentations were performed in Tractome, which integrates clustering algorithms with the visual inspection and manipulation of streamlines. Shape analysis was conducted on each bundle. A paired *t*-test was performed on the irregularity measurement to compare segmentations achieved with the two approaches. Qualitative differences were evaluated through visual inspection. Streamline-based segmentation consistently yielded significantly lower irregularity scores (*p* < 0.001) compared to ROI-based segmentation for all the examined bundles, indicating more compact and accurate bundle reconstructions. Qualitative assessment identified common biases in ROI-based segmentations, such as the inclusion of anatomically implausible streamlines or streamlines with undesired trajectories. Streamline-based bundle segmentation with Tractome provides reliable and more accurate reconstructions compared to the ROI-based approach. By directly manipulating streamlines rather than relying on voxel-based ROI delineations, Tractome allows us to discern and discard implausible or undesired streamlines and to identify the course of WM bundles even when the anatomy is distorted by the lesion. These features make Tractome a robust tool for bundle segmentation in clinical contexts.

## 1. Introduction

Over the past decade, substantial evidence has established a significant correlation between the extent of resection (EOR) and the overall survival (OS) in patients affected by both low- and high-grade gliomas (LGGs and HGGs, respectively) [1,2]. Notably, resecting the infiltrating and non-enhancing component of the tumor positively impacts the OS even in HGGs [3,4,5], leading to the emergence of the concept of supra-total resection (SpTR) in neurosurgical practice. SpTR must be performed with due consideration for the preservation of the neurological and cognitive status of the patients, ultimately aiming to enhance their quality of life [6].

The current understanding of the mechanisms underlying cognitive functions points to the importance of white matter (WM) connections [7] and suggests the relevance of preserving functionally essential WM bundles during tumor resection. Awake surgery with direct electrical stimulation (DES) stands as the gold standard for defining the functional subcortical boundaries of resection, particularly when mapping language and higher cognitive functions [8,9]. Similarly, intraoperative monitoring during asleep procedures, including somatosensory and motor evoked potentials (SSEP and MEP), has proven crucial, particularly in lesions affecting motor areas or the pyramidal tract [10,11]. DES and intraoperative monitoring have provided valuable insights into the functional role of the human WM [12,13,14], contributing to the delineation of a minimal core set of WM pathways to be considered when planning the safe resection of lesions infiltrating the WM. However, precise knowledge of the structural WM anatomy at the individual level is required to enhance the value of functional intraoperative mapping, especially in cases of large lesions that alter the natural expected course of WM fibers.

Diffusion weighted imaging (DWI)-based tractography holds inestimable value in providing a unique glance into the structural organization of the WM of our patients in vivo, and connections known to be functionally critical can be identified through the process of bundle segmentation. The analysis of the WM pathways of the single patient yields useful information on the relationships between WM bundles and tumors, including the localization of the main portion of the bundles, the spatial displacement of fibers due to the lesion, and the anatomical relationships among the different affected bundles. Tractography is an excellent tool for building precise virtual models of the subcortical anatomy in pathological cases. However, a significant challenge for clinicians lies in the lack of standardized protocols for extracting WM bundles from a whole-brain tractogram [15], which is related to the ongoing debate surrounding the definition of many WM bundles [16].

The first study of bundle segmentation on a set of virtual fiber tracks [17] established a common practice for this procedure. It involves delineating regions of interest (ROIs) based on prior anatomical or functional knowledge to define the waypoints or terminal regions of the WM tract under investigation [18,19]. This process entails drawing a volume on coronal, sagittal, or axial projections by selecting voxels from the anatomical MR image that the streamlines (i.e., virtual reconstructions of WM fibers) of the target bundle are expected to—or should not—cross. While the ROI-based approach remains the most widely adopted strategy for virtual bundle segmentation, it is not the only method available. Streamline-based approaches typically use clustering algorithms to group streamlines with similar shapes and closely related courses [20,21]. This aligns with the traditional anatomical process of bundle recognition and classification, which primarily relies on identifying the characteristic macroscopically visible trajectory of WM fibers. As opposed to ROI-based segmentation, streamline-based approaches enable operators to directly act on the streamlines, bypassing intermediary means such as voxels of anatomical images. By doing so, streamline-based approaches address the inherent paradox of ROI-based approaches, where segmentation is performed without direct consideration of the fiber reconstructions themselves.

Herein, we qualitatively and quantitatively compare the segmentation of four widely recognized association WM bundles using both ROI-based and streamline-based segmentation approaches. Considering the importance of visual inspection in clinical practice, we perform ROI-based segmentations with the visualization software Trackvis [22], and streamline-based segmentations with the software tool Tractome [23], which leverages clustering algorithms while providing an interface for visual inspection.

## 2. Material and Methods

### 2.1. Imaging Acquisition, Processing, and Tractography

The standard protocol for preoperative planning at ‘Santa Chiara’ University-Hospital in Trento, Italy, involves MRI acquisition on a 1.5 T GE Clinical Optima MR450 scanner (GE Healthcare, Milwaukee, WI, United States). The present study includes preoperative T1-weighted volumetric images (axial acquisition; TR/TI/TE: 10.64/450/4.23 ms; FA: 12°; square FOV: 256 mm; voxel size: 1 × 1 × 1 mm^3^) and DW images (single-shot multislice SE-EPI sequence; b = 0 s/mm^2^; 60 directions at b = 1000 s/mm^2^; axial acquisition; TR/TI/TE: 13,000/89/95.8 ms; FA: 90°; square FOV: 256 × 256 mm, voxel size: 2.4 × 2.4 × 2.4 mm^3^) from 25 HGG patients scheduled for tumor resection. Data usage was approved by the local ethics committee (ID: A734). Following the pipeline described in [24], DW images were preprocessed for artifacts correction and deterministic constrained spherical deconvolution (CSD) tractography was performed. Color-coded fractional anisotropy (cFA) maps (i.e., directionally encoded color maps [25]) were generated. To facilitate smooth data visualization, whole-brain tractograms were resampled to 1,000,000 streamlines.

### 2.2. Bundle Segmentation

The segmentation of four association WM bundles was performed in the healthy hemisphere of each patient by an expert anatomist (L.V.) using both ROI- and streamline-based approaches. The segmentations were verified by a second expert anatomist (L.Z.). The WM bundles analyzed in the study include the following:
-The arcuate fasciculus (AF) [12]. This WM bundle is composed of the fronto-temporal fibers of the Superior Longitudinal System (SLS) [16,26].-The indirect anterior segment of the AF [27], also referred to as the third segment of the superior longitudinal fasciculus (SLF III) [28]. This bundle comprises the most ventral and lateral fibers of the fronto-parietal SLS [16,26].-The indirect posterior segment of the AF [27]. This WM bundle includes the most anterior set of fibers of the Posterior Transverse System (PTS) connecting the temporal and the parietal cortices [26].-The inferior fronto-occipital fasciculus (IFOF) [29]. This bundle comprises long fibers of the Inferior Longitudinal System (ILS) connecting the frontal to the occipital, temporal, and parietal cortices. These fibers converge in a stem located at the level of the anterior floor of the external/extreme capsule [26]. Unlike the short component of the ILS, which bends into the frontal pole after passing the level of the stem, they follow a more longitudinal course.

Both streamline- and ROI-based approaches rely on visual inspection for the segmentation of the WM bundles. While this introduces a certain degree of subjectivity, it remains the most reliable method available for achieving patient-specific segmentations, particularly in clinical settings where automated techniques cannot yet account for complex pathological anatomies [30].

#### 2.2.1. ROI-Based Segmentation

ROI-based segmentation was conducted using the software tool TrackVis (V0.6.2.1) [22]. Given the associational nature of the bundles considered in this study, a slice filter corresponding to the midline was systematically used to exclude commissural fibers. ROIs were manually delineated on the cFA of the patients following specific heuristics, which were defined to establish a systematic and reproducible workflow for the virtual segmentation of the WM bundles (Figure 1) as follows:
-The segmentation of the AF involves two inclusion ROIs. The first ROI is drawn on a coronal slice at the level of the deep WM just posterior to the central sulcus. This area can be identified as a bright green triangle on the cFA coronal projection, which indicates the main antero-posterior course of the longitudinal portion of the AF. After bending around the Sylvian fissure, the fibers of the AF take a dorso-ventral orientation and reach the temporal lobe. To catch this characteristic course, a second ROI is drawn on the axial plane of the cFA at the level of the deep WM below the posterior part of the superior temporal gyrus, where a blue triangle can be identified. The AF is therefore extracted from each patient’s whole-brain tractogram by setting these two drawn volumes as ROI filters, supporting the logical operator ‘Any Part’. The ROIs establish the two waypoints through which fibers must pass in order to be considered as part of the AF (Figure 1A).-The indirect anterior segment of the AF is extracted starting from the same two ROIs as for the segmentation of the AF. The coronal ROI remains unchanged and serves as an inclusion waypoint ROI (i.e., set as an ROI filter supporting the logical operator ‘Any Part’), while the axial ROI is enlarged to obtain a plane segregating the temporal and the parietal lobes. This plane is used as an exclusion ROI (i.e., set as an ROI filter supporting the logical operator ‘No Part’), to support the exclusion of fibers bending into the temporal cortex after arching around the Sylvian fissure (Figure 1B).-The indirect posterior segment of the AF is extracted starting from the same two ROIs as for the segmentation of the AF. The axial ROI serves as an inclusion waypoint ROI (i.e., set as an ROI filter supporting the logical operator ‘Any Part’), while the coronal ROI is expanded to obtain a plane segregating the frontal and the parietal lobes. This plane is used as an exclusion ROI (i.e., set as an ROI filter supporting the logical operator ‘No Part’), to support the exclusion of all the fibers originating in the frontal cortex (Figure 1C).-The IFOF is segmented using a stem-based approach as described in [29]. An inclusion waypoint ROI (i.e., set as an ROI filter supporting the logical operator ‘Any Part’) is drawn on the coronal plane at the level of the external/extreme capsule. This region through which all the fibers of the ILS are conveyed to pass can be identified as a small green rectangle, indicating the main antero-posterior orientation of the fibers at that level. Two exclusion ROIs (i.e., set as an ROI filter supporting the logical operator ‘No Part’) are drawn above and below the stem ROI to exclude ILS fibers from bending ventrally and looping back towards the temporal pole (i.e., the uncinate fasciculus) and artefactual fibers looping back dorsally into the frontal lobe, respectively (Figure 1D).

In standard clinical settings, the manual ROI-based segmentation of WM bundles is fine-tuned by applying additional exclusion ROIs to further eliminate streamlines that the operator judges as anatomically implausible or not representative of the bundle of interest. The number of additional exclusion ROIs can be virtually infinite, and their location varies based on the single case examined, introducing further variability in the ROI-based segmentation of the WM bundles. After initially segmenting the bundles following the heuristics defined above, the reconstruction of the AF and the IFOF still included several artefactual streamlines. By visually inspecting the obtained bundles and analyzing the main patterns leading to unsatisfactory representations, we systematically modified existing exclusion ROIs or added new ones in all patients to improve AF and IFOF reconstructions while still adhering to well-defined segmentation heuristics consistent across all the patients. Specifically, an exclusion ROI was drawn on the axial and coronal planes at the level of the insula for each AF segmentation. This ROI was crucial for eliminating artefactual streamlines that resemble the course of the AF but bend inferiorly with a vertical trajectory before ending in the frontal cortex. This common artifact is due to the tracking algorithm being misled by the direction of more medial projection fibers. For the IFOF, the two exclusion ROIs initially drawn on the coronal plane were extended to the axial plane, similar to the exclusion ROI just described for the AF. This extension allows the dorsal exclusion ROI, initially designed to exclude fibers that, after coursing towards posterior areas of the brain, bend back dorsally and terminate in the frontal cortex, to also eliminate more medial projection-like fibers with a vertical course. The extension of the inferior coronal exclusion ROI helps discard streamlines that, after coursing posteriorly towards the occipital lobe, loop back and follow the direction of the inferior longitudinal fasciculus (ILF) to then terminate in the temporal lobe just before the temporal pole. These ROIs are depicted in Figure 1 as part of our ROI-based segmentation heuristics.

#### 2.2.2. Streamline-Based Segmentation

Streamline-based segmentation was performed using Tractome (http://tractome.org, accessed on 1 February 2024) [23], a software tool that supports an interactive process interleaving steps of fast clustering and steps of visual inspection and manipulation of streamlines. Similarly to TrackVis, bundle segmentation in Tractome begins with a whole-brain tractogram. Upon opening the tractogram in the software, streamlines undergo clustering, displaying a series of prototypes overlaid on the anatomical image provided. Prototypes are streamlines representative of groups of fibers with similar shapes and close spatial relationships. They provide an abstract simplification of the tractogram, enabling user interaction. Indeed, interaction with the entire set of streamlines of a whole-brain tractogram would not be viable due to the high number of streamlines that it contains. The interaction is supported by a blended mouse and keyboard control interface: while the mouse pointer defines which prototype the user is working on, a series of actions can be initiated by pressing specific keys on the keyboard. The segmentation starts with the identification and selection of the prototypes representing the bundle of interest and the disposal of irrelevant ones. Streamline-based bundle segmentation with Tractome is an iterative process. The user can adjust the number of clusters to recursively adapt the representation of the remaining set of streamlines. As the selection and elimination of irrelevant clusters progress towards the desired segmentation, the number of clusters can be increased until each individual streamline is represented and can be directly selected. This approach allows users to identify the bundle by considering the shape of the streamlines themselves, without the need to annotate MR images for indirect streamlines selection as required in ROI-based approaches. Figure 2 illustrates the process of streamline-based bundle segmentation with Tractome for the four bundles considered in this study.

### 2.3. Quantitative and Qualitative Analyses

Shape analysis [31] was conducted to describe each segmented bundle through the SCILPY script ‘scil_bundle_shape_measures.py’ (https://github.com/scilus/scilpy accessed on 1 May 2024). Among the several metrics proposed in this framework, the irregularity measurement defines the compactness of a bundle, factoring in its surface area, diameter, and length. This index is sensitive to the presence of streamlines deviating from the overall main course of the bundle, with lower irregularity values indicating more compact bundles. We therefore selected the irregularity measurement to compare the segmentation of the same bundle in the same patient achieved through ROI- and streamline-based approaches. Given the normal distribution of the data (*p* > 0.05 for the Shapiro–Wilk test), parametric testing was performed [32]. We performed a paired *t*-test with the SciPy ‘ttest_rel’ function [33] to test for significant differences in the mean irregularity of the two related samples (ROI- vs. streamline-based segmentation). We calculated Cohen’s d to quantify the effect size [34]. Analyses were performed in Python 3.10, and the significance level for the paired *t*-test was set at *p* < 0.05. A qualitative assessment of the differences between the two segmentation strategies applied to the extraction of the same bundle was conducted by a third expert anatomist (S.S) through visual inspection.

## 3. Results

In this study, we assessed the applicability and effectiveness of two different approaches for bundle segmentation, namely ROI- and streamline-based segmentation, on a clinical dataset of 25 HGG patients. The focus was on four well-known association WM bundles of the human brain: the AF, the indirect anterior and the posterior segments of the AF, and the IFOF. Segmentations were performed in the healthy hemisphere. A numeric description of each segmented bundle based on the Shape Analysis [31] is provided in Appendix A.

### 3.1. Quantitative Analysis

To evaluate possible differences in the results of bundle segmentation based on the two approaches, we considered the irregularity measurement [31]. Lower irregularity values indicate more compact segmentations, with fewer streamlines deviating from the main course of the bundle. Table 1 reports the descriptive statistics on the irregularity measurement for each bundle and each segmentation approach.

We conducted a paired *t*-test to determine significant differences in the mean irregularity between ROI- and streamline- based segmentation for each bundle (i.e., AF, indirect anterior and posterior segments of the AF, and IFOF). As shown in Figure 3 and reported in Table 2, the results indicated statistically significant differences for all the bundles with *p* < 0.001 and a large effect size as described by Cohen’s d.

### 3.2. Qualitative Assessment

Once it was assessed that ROI- and streamline-based segmentations are significantly different, we conducted a visual inspection to describe the macroscopic variations in the bundle segmentation results. Overall, bundles segmented through a ROI-based approach exhibit more individual streamlines or small groups of streamlines deviating from the main course of the bundle, often influenced by the directionality of nearby fiber populations. Specifically, for each segmented bundle, we identified consistent patterns of undesired streamlines across the investigated population (Figure 4).

The ROI-based segmentation of the AF was systematically characterized by streamlines ending in the parietal lobe. These fibers, by definition, do not belong to the AF, as a long connection directly linking the frontal and the temporal cortices. Given the mandatory requirement of crossing both the coronal parietal and the axial temporal waypoint ROIs, these streamlines also do not belong to the indirect anterior or posterior segments of the AF. They have an anatomically implausible course and should therefore be discarded (Figure 4A). The ROI-based segmentations of the indirect posterior and anterior components of the AF are characterized by short U-shaped fibers, which should be excluded since they do not pertain to these long WM fiber bundles. In particular, the inclusion of these U-shaped fibers in the reconstruction of the bundles depends heavily on the positioning of the inclusion ROI. Indeed, using a sole waypoint ROI captures the entire cortical extension of the bundle without imposing any prior on its termination territories, but it can lead to the inclusion of these shorter connections (Figure 4B,C). Finally, the ROI-based segmentations of the IFOF include arched streamlines that, after leaving the frontal cortex and passing through the stem of the bundle, take a smooth turn, looping back dorsally towards the frontal cortex, following the main directionality of the AF (Figure 4D).

Although some of these artefactual streamlines in the ROI-based segmentations can be removed by adding further exclusion ROIs, it becomes complicated when these streamlines are not completely isolated from the bundle of interest but merge with it. Indeed, the selection of one voxel includes all the fibers that pass through that voxel, not allowing it to distinguish between the fibers to preserve and those to discard.

## 4. Discussion

DWI-based tractography enables the reconstruction of the WM pathways of the brain, and bundle segmentation allows for the extraction of specific sets of functionally relevant fibers that organize into distinct wiring patterns. An accurate and anatomically reliable reconstruction of WM bundles is particularly crucial in clinical and surgical settings, especially for preoperative planning and intraoperative navigation during tumor resection. In this study, we compare two different approaches for bundle segmentation, contrasting the results obtained with the widely used ROI-based approach with those achieved through a streamline-based approach in a clinical population of 25 HGG patients. We examined the AF, the indirect posterior and the anterior segments of the AF, and the IFOF. By considering the irregularity measurement as an index of the compactness of a bundle, we found that streamline-based bundle segmentation consistently yields significantly lower irregularity scores compared to the ROI-based approach, indicating more compact and neat segmentations. Through visual inspection, we identified common patterns of anatomical implausible streamlines that characterize the ROI-based segmentation of each WM bundle, contributing to systematically higher irregularity values.

The increased presence of anatomically implausible streamlines or streamlines deviating from the main course of the bundle in ROI-based compared to streamline-based segmentations stems from inherent features of the two approaches. The classical ROI-based approach consists of an indirect selection process, where fibers of interest are isolated from a whole-brain tractogram based on the expected termination territories or waypoints of the bundle. In software programs like TrackVis, ROIs are manually drawn on anatomical images, and the effects of the ROI seconds their application to the tractogram. Conversely, streamline-based bundle segmentation in Tractome allows for direct interaction with the streamlines, enabling the user to iteratively cluster, examine, discard, or retain connectivity patterns based on their course, terminations, and spatial relationships. The streamline-based approach, implemented in a user-friendly interface in Tractome, offers significant advantages compared to ROI-based approaches. Since one single voxel may be crossed by multiple streamlines from different fiber populations [35], acting on the voxel does not allow for the distinction between those that are of interest and irrelevant ones. The inclusion of fibers not belonging to the bundle of interest in ROI-based approaches can only be mitigated by delineating additional exclusion ROIs or reducing the number of voxels of the inclusion ROI. This entails the risk of discarding large amounts of plausible streamlines, and therefore relevant anatomical information, throughout the process. Moreover, it introduces further variability in the segmentation results, undermining reproducibility and complicating the creation of bundle-tailored systematized protocols for segmentation.

Predefined segmentation protocols are mandatory for establishing the anatomical constraints that identify a bundle in ROI-based segmentations. However, the definition of many WM tracts remains under debate [16], and no definitive protocol exists for determining which type of ROI must be placed, and where it should be placed, to extract a specific bundle [15]. The process of ROI placement is prone to high variability, not only across different operators but also within the same operator when blindly repeating the segmentation on the same dataset [36]. Even when a fictitious standardized segmentation protocol is provided, manual ROI drawing remains susceptible to operator-dependent variability [36,37].

In this work, we chose to perform bundle segmentation in the healthy hemisphere of the patients. This enabled the evaluation of the reliability of the streamline-based approach compared to the ROI-based approach, circumventing the debate on the anatomical reliability of bundles deformed by lesions while still testing its performance on data acquired with clinical parameters. By using clinical standard dMRI data, we ensure that our findings are directly applicable to clinical data, whose quality is often influenced by the constraints of clinical protocols such as acquisition time, resolution, and hardware limitations. Both the quantitative and qualitative results of this study support the specificity and accuracy of the reconstructions obtained with the interactive streamline-based approach supported by Tractome. We claim that the main positive features of bundle segmentation with Tractome discussed above are particularly relevant in pathological settings and carry additional advantages when the WM anatomy is significantly deformed by lesions. First, tractography clustering with the display of representative prototypes allows clinicians to assess how the WM anatomy of the single patient may be altered by the lesion (e.g., if the WM bundles have been pushed more medially, laterally, dorsally, or ventrally, or if there are fibers embracing the lesions). This approach also prevents the need to make assumptions about the location of landmarks typically used for identifying the waypoints of the bundle, which may no longer apply in the presence of a lesion. Furthermore, it enables the user to select and discard implausible streamlines individually, without affecting plausible ones—something that is more challenging in ROI-based approaches where voxel selection indistinguishably impacts all the streamlines crossing that voxel. This feature responds to the need for anatomically reliable and exhaustive representations of the extension of WM bundles for preoperative planning and intraoperative neuronavigation. Finally, direct and simplified visualization of the WM anatomy, as well as dynamic interaction with tractography data supported by Tractome, align with the expertise of clinicians and neurosurgeons, who are increasingly skilled in human WM anatomy through microdissection training. As we demonstrate the methodological strengths of a streamline-based approach for bundle segmentation on standard clinical data, we set the basis for future investigations and clinical applications. Future studies are needed to apply our comparative approach to lesioned hemispheres, and to investigate whether the use of bundle segmentations obtained with streamline-based approaches carries clinical benefits, such as a better prediction of the clinical outcome of patients compared to ROI-based segmentations.

## 5. Limitations

We acknowledge that the sample size of this study is relatively small, which may limit the generalizability of our findings. Despite this limitation, the significant differences observed between the ROI- and streamline-based approaches, supported by the calculation of Cohen’s d for effect size, provide robust evidence for the reliability of our findings. Future studies with larger and more diverse cohorts will be critical to further validate and extend the applicability of our conclusions to other clinical and research contexts. Additionally, this study was conducted on the healthy hemisphere of the patients we selected. This choice was aimed to test and describe the differences in bundle segmentations that can be achieved via ROI- and streamline-based approaches in optimal settings, setting aside the debate related to what is the ground truth of anatomical distortions in the presence of a lesion, while still using data acquired with clinical standards. Future works including hemispheres affected by lesions are needed to better demonstrate the potential benefits of streamline-based approaches for bundle segmentation in distorted anatomies.

## 6. Conclusions

The present study demonstrates that streamline-based bundle segmentation, as implemented in the software Tractome, provides more compact and anatomically reliable representations of WM bundles compared to the traditional ROI-based approach, based on quantitative irregularity scores and the qualitative assessment of streamlines plausibility. These findings highlight methodological advantages that may support clinicians in visualizing and interpreting WM anatomy, particularly in the presence of lesions that deform the anatomy of the brain. Future studies will be required to validate these findings across broader datasets and assess their impact on different clinical scenarios.

## Figures and Tables

**Figure 1 brainsci-14-01232-f001:**
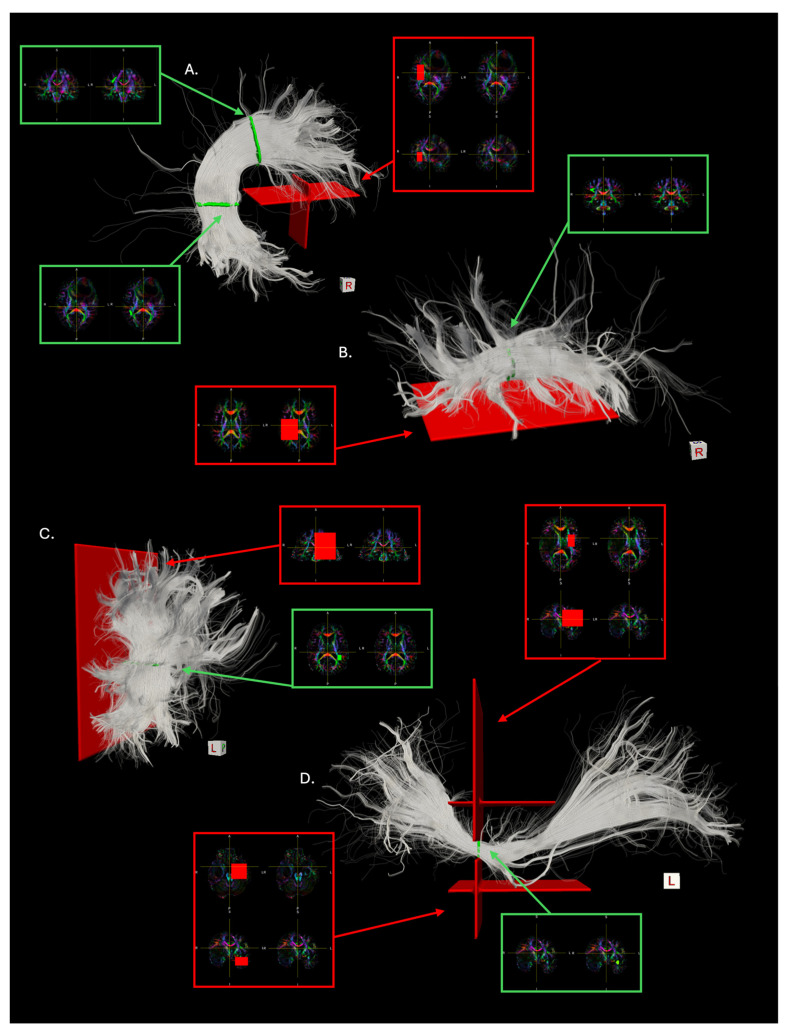
The ROI placement for the extraction of the (**A**) AF, (**B**) indirect anterior segment of the AF, (**C**) indirect posterior segment of the AF, and (**D**) IFOF according to the heuristics described for ROI-based segmentation. The 3D rendering in each panel shows the reconstructed WM fibers (in transparency) and the respective ROIs. The 2D inset panels display the cFA of the patient on which the ROIs were drawn, with and without ROI overlay. Green indicates inclusion ROIs, while red indicates exclusion ROIs.

**Figure 2 brainsci-14-01232-f002:**
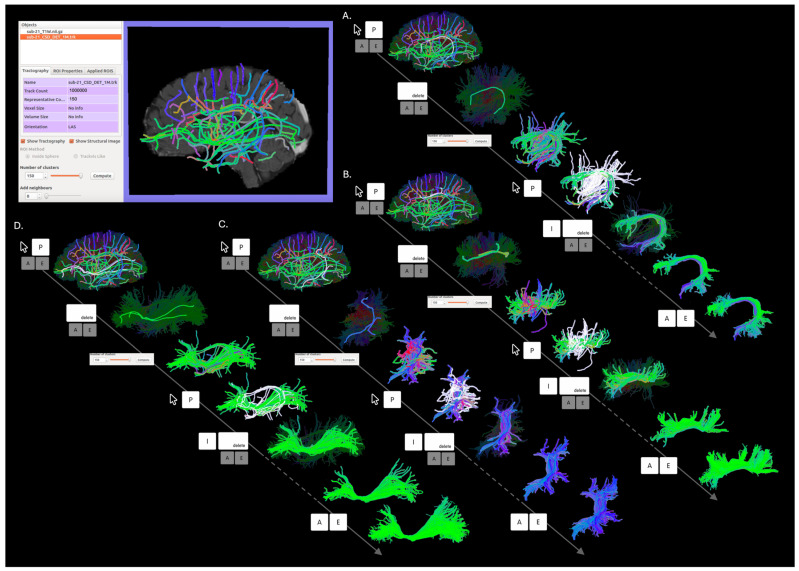
Streamline-based segmentation of the (**A**) AF, (**B**) indirect anterior segment of the AF, (**C**) indirect posterior segment of the AF, and (**D**) IFOF in Tractome. The top left quadrant represents the main view in Tractome: upon the opening of a whole-brain tractogram, clustering is automatically performed, and prototypes representing the main connectivity patterns are shown. The pictures along each arrow represent the different steps we performed to achieve the segmentation of the four bundles from the same patient, from the selection of their initial representative clusters, their cleaning, and the re-clustering to further improve the segmentation (iteration of the different steps is represented by the dotted line), to their final representation. Below each line, the keyboard keys, the mouse cursor, and the button for cluster re-computing define the different segmentation steps performed (P = pick representative, I = invert selection, backspace = remove unselected, A = select all representatives, E = expand selection).

**Figure 3 brainsci-14-01232-f003:**
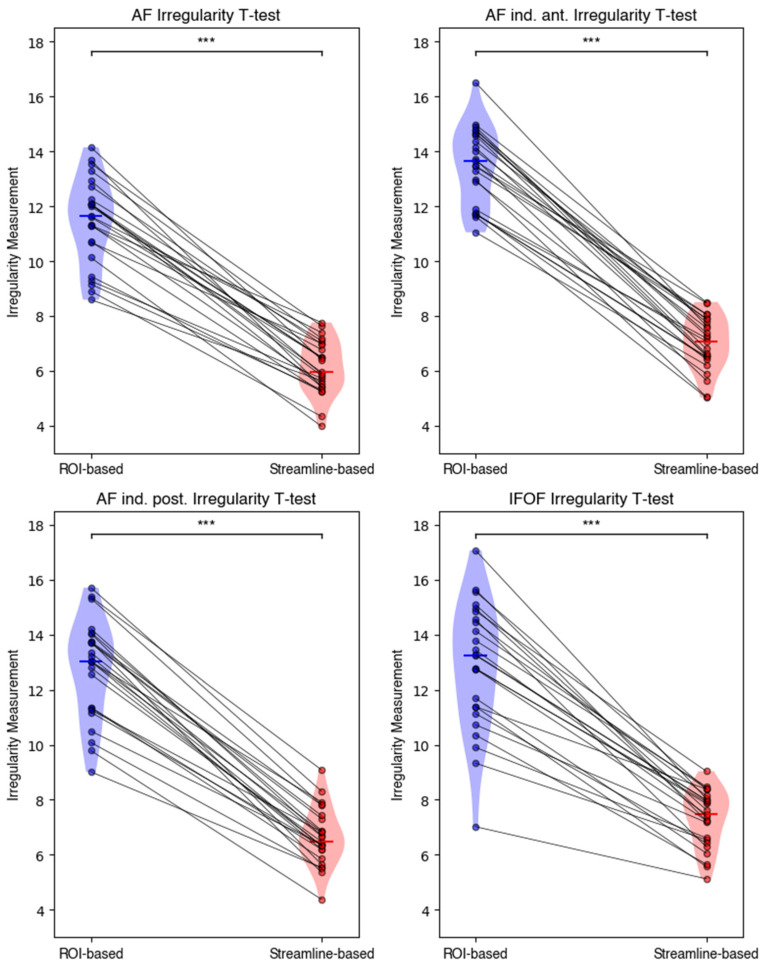
The violin plots display the distribution of the irregularity measurements for each WM bundle across the 25 HGG patients segmented according to ROI- (blue) and streamline-based (red) approaches. The plots correspond to the four different WM bundles considered in the study: the AF, the indirect anterior segment of the AF, the indirect posterior segment of the AF, and the IFOF. Each dot represents the irregularity measurement of a single patient, while gray lines connect the measurements from the same patients across the two categories representing the segmentation approach. The significantly lower irregularity scores observed for the streamline-based segmentations compared to the ROI-based ones across all bundles (*** *p* < 0.001) suggests that the streamline-based segmentations achieved with Tractome are more compact compared to the traditional ROI-based segmentation approaches. Statistical significance was determined using a paired *t*-test.

**Figure 4 brainsci-14-01232-f004:**
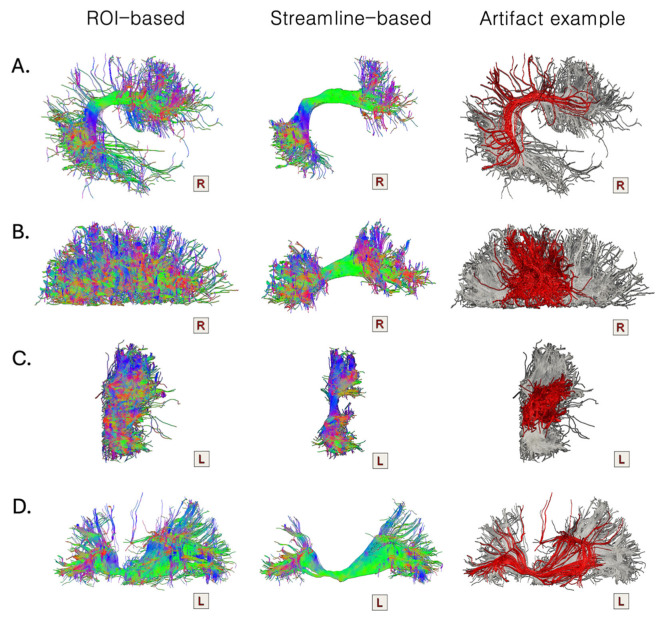
A qualitative comparison of the macroanatomical differences between the ROI- and streamline-based segmentations applied to the extraction of the same bundle ((**A**) the AF, (**B**) the indirect anterior segment of the AF, (**C**) the indirect posterior segment of the AF, and (**D**) the IFOF) from the same initial whole-brain tractogram. Overall, the ROI-based segmentations (first column) appear ‘messier’ compared to the streamline-based ones (second column). Artefactual streamlines or streamlines not belonging to the bundle of interest cannot be removed with ROI-based segmentation approaches while preserving the overall integrity of the bundle. Recurrent patterns of artefactual streamlines that can be identified in the ROI-based segmentations and that can be removed with the streamline-based segmentation in Tractome are reported in the third column in red, overlaid on the original ROI-based segmentation.

**Table 1 brainsci-14-01232-t001:** Descriptive statistics on irregularity measurement for each bundle of interest segmented with ROI- and streamline-based approaches. We report mean, standard deviation (std), minimum value (min), 25th percentile (25%), 50th percentile (50%), 75th percentile (75%), and maximum value (max). AF, arcuate fasciculus; IFOF, inferior fronto-occipital fasciculus.

Bundle	Segmentation Approach	Mean	Std	Min	25%	50%	75%	Max
AF	ROI-based	11.475	1.555	8.592	10.683	11.629	12.241	14.147
	Streamline-based	6.129	0.978	3.977	5.423	5.945	6.970	7.764
AF ind. ant.	ROI-based	13.488	1.404	11.050	11.885	13.654	14.653	16.505
	Streamline-based	6.989	0.971	5.030	6.491	7.076	7.776	8.509
AF ind. post.	ROI-based	12.762	1.778	9.014	11.344	13.033	13.741	15.708
	Streamline-based	6.646	1.041	4.377	6.199	6.472	7.310	9.086
IFOF	ROI-based	12.847	2.321	7.019	11.384	13.238	14.544	17.069
	Streamline-based	7.326	1.038	5.125	6.550	7.465	8.044	9.040

**Table 2 brainsci-14-01232-t002:** The results of the paired *t*-test conducted on each bundle category to compare the segmentations achieved via the ROI- and streamline-based approaches. We report the t-statistics, the *p*-values (all *p*-value < 0.001), and Cohen’s d. AF, arcuate fasciculus; IFOF, inferior fronto-occipital fasciculus.

Bundle	*t*-Statistics	*p*-Value	Cohen’s d
AF	19.862	0.00000000000000020955	3.972
AF ind. ant.	25.372	0.00000000000000000076	5.074
AF ind. post.	23.904	0.00000000000000000302	4.781
IFOF	15.322	0.00000000000006828710	3.064

## Data Availability

The data presented in this study are available upon reasonable request from the corresponding author due to their clinical nature.

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
