# Peer review of "Changing the Paradigm for Tractography Segmentation in Neurosurgery: Validation of a Streamline-Based Approach"

_brainsci, 2024, doi:10.3390/brainsci14121232_

Round 1

Reviewer 1 Report

Comments and Suggestions for Authors

Although the study offers a comparison between ROI-based and streamline-based bundle segmentation method, there are several concerns that need to be addressed.

1.The sample size is small, potentially limiting the generalizability of findings.

2. The streamline-based method's reliance on visual inspection, which makes it subjective.

3. The issue of computional complexity and time consumption of the two methods is not discussed in the study.

4. A more in-depth discussion on the clinical implications of the findings is needed to assess their practical significance.

5. The quality of dMRI data has a considerable influence on tracking results. Authors should also take into consideration this factor.

Author Response

We thank Reviewer 1 for taking the time to review this manuscript. Please find the detailed responses below and the corresponding revisions highlighted in red in the re-submitted file.

Comment 1: The sample size is small, potentially limiting the generalizability of findings.

Response 1: We appreciate the observation of Reviewer 1 regarding the sample size of our study. While we acknowledge that the sample size is relatively small, it is consistent with other clinical studies performing tractography bundle segmentation on tumor patients (e.g., O’Donnel et al., 2017; Zoli et al., 2021; Fekonja et al., 2019). Following this comment and the suggestion of Reviewer 2 in comment #4, we report the calculation of Cohen’s d (Cohen, 1969) to quantify the effect size of the paired t-test in Table 2, in the Methods (Line 266) and in the Results section (Line 293). To address concerns about generalizability of findings, we have included the following acknowledgement in the Limitations section (Lines 427-432): “We acknowledge that the sample size of this study is relatively small, which may limit the generalizability of our findings. Despite this limitation, the significant differences observed between the ROI- and streamline-based approaches, supported by the calculation of Cohen’s d for effect size, provide robust evidence for the reliability of our findings. Future studies with larger and more diverse cohorts will be critical to further validate and extend the applicability of our conclusions to other clinical and research contexts. ” 

Comment 2: The streamline-based method's reliance on visual inspection, which makes it subjective.

Response 2: We thank Reviewer 1 for highlighting the reliance of the streamline-based approach on visual inspection, which may introduce subjectivity during cluster selection. While we acknowledge this limitation, it is important to note that manual segmentation of WM bundles in general relies heavily on visual inspection (Zigiotto et al., 2022; Forkel et al., 2023; Fekonja et al., 2019), and this is the reason why we introduced here also a quantitative method of analysis of the results, in respect to previous studies. Regarding the visual inspection, this includes also the ROI-based approaches, currently considered the gold standard for the segmentation of WM bundles in clinical settings. Visual inspection remains the most reliable means of achieving and evaluating patient-specific tractography segmentations, given the limitations of automated methods in accounting for individual anatomical variability especially in pathology (Bertò et al., 2021; Ghazi et al., 2023). We argue that the streamline-based approach facilitates inspecting and discarding anatomically implausible streamlines, which can be challenging to detect in ROI-based approaches. This is particularly true for probabilistic constrained-spherical deconvolution tractography, as used in the present work, where plausible and implausible streamlines may be indiscernible at the voxel level (Tournier et al., 2007). By providing an interactive clustering framework, streamline-based segmentation in Tractome facilitates the selection and disposal of implausible streamlines, reducing the likelihood of retaining them while preserving anatomically plausible ones. While the advantages of adopting a streamline- over a ROI-based method for the identification of implausible streamlines upon visual inspection have been already largely discussed in the Discussion section of the manuscript, thanks to this comment we now added to the Methods section a clarification regarding the reliance of both streamlines- and ROI-based approaches on visual inspection as follows (Lines 140-144): “Both streamline- and ROI-based approaches rely on visual inspection for the segmentation of WM bundles. While this introduces a certain degree of subjectivity, it remains the most reliable method available for achieving patient-specific segmentations, particularly in clinical settings where automated techniques cannot yet account for complex pathological anatomies.”

Comment 3: The issue of computional complexity and time consumption of the two methods is not discussed in the study.

Response 3: We thank Reviewer 1 for raising this observation. In this study, the computational complexity to obtain the tractogram was not emphasized because the critical factor for ROI- and streamline-based approaches lies in the manual identification and refinement of WM bundles rather than in the computational processes themselves. Indeed, both ROI- and streamline-based approaches rely on already processed tractography data, with bundle segmentation performed as a separate step (Jeurissen et al., 2019; Forkel et al., 2023). Thus, the primary challenge is not the computational load of data processing but the time and expertise required for the manual steps, including the placement of ROIs in the ROI-based approach or the selection of clusters in the streamline-based approach. 

Comment 4: A more in-depth discussion on the clinical implications of the findings is needed to assess their practical significance.

Response 4: We thank Reviewer 1 for the valuable suggestion to expand on the future clinical implications of our findings. The primary objective of our study was to demonstrate in a clinical-like setting (i.e., using data acquired according to standard and common clinical imaging parameters) the main advantages of the streamline- compared to the ROI-based approach, highlighting those features that make the streamline-based approach relevant in the case of pathological lesions that deform the anatomy of the brain. This study highlights how the streamline-based approach facilitates direct and simplified visualization of WM anatomy through clustering, as well as dynamic interaction with tractography data, therefore providing a suitable platform for the exploration of pathological anatomies. It enables the user to select and discard implausible streamlines individually, without affecting plausible ones - something that is more challenging in ROI-based approaches, where voxel selection indistinguishably impacts all the streamlines crossing that voxel. This feature responds to the need of anatomically reliable and exhaustive representations of the extension of WM bundles for preoperative planning and intraoperative neuronavigation. We further emphasize that bundle segmentation with a ROI-based approach, which relies on termination or waypoints ROI, can be more vulnerable to the alterations of brain morphology in pathological conditions. This makes the ROI-based approach more prone to errors and misclassifications in the segmentation results as anatomical landmarks can be lost or significantly vary in the case of brain lesions. As suggested, we have now implemented in the manuscript a more in-depth discussion regarding the aim of the study and the clinical expendability of our result. The revised statement reads as follows (Lines 401-424): “We claim that the main positive features of bundle segmentation with Tractome discussed above are particularly relevant in pathological settings, and carry additional advantages when the WM anatomy is significantly deformed by lesions. First, tractography clustering with the display of representative prototypes allows clinicians to assess how the WM anatomy of the single patient may be altered by the lesion (e.g., if WM bundles have been pushed more medially, laterally, dorsally or ventrally, or if there are fibers embracing the lesions). This approach also prevents the need to make assumptions about the location of landmarks typically used for identifying the waypoints of the bundle, which may no longer apply in the presence of a lesion. Furthermore, it enables the user to select and discard implausible streamlines individually, without affecting plausible ones - something that is more challenging in ROI-based approaches, where voxel selection indistinguishably impacts all the streamlines crossing that voxel. This feature responds to the need for anatomically reliable and exhaustive representations of the extension of WM bundles for preoperative planning and intraoperative neuronavigation. Finally, direct and simplified visualization of the WM anatomy, as well as dynamic interaction with the tractography data supported by Tractome, align with the expertise of clinicians and neurosurgeons, who are increasingly skilled in human WM anatomy through microdissection training. As we demonstrate the methodological strengths of a streamline-based approach for bundle segmentation on standard clinical data, we set the basis for future investigations and clinical applications. Future studies are needed to apply our comparative approach to lesioned hemispheres, and to investigate whether the use of bundle segmentations obtained with streamline-based approaches carries clinical benefits, such as a better prediction of the clinical outcome of patients, compared to ROI-based segmentations.”

Comment 5: The quality of dMRI data has a considerable influence on tracking results. Authors should also take into consideration this factor.

Response 5: We thank Reviewer 1 for stressing that dMRI data quality influences tractography results. We fully acknowledge that the quality of dMRI data can significantly impact the accuracy and reliability of tractography results (Thomas et al., 2014). To address this concern, we intentionally designed our study to use data acquired under standard clinical conditions commonly employed in routine clinical practice, and the same dataset for both the techniques we compared in this study. This decision was made to ensure that our findings are directly applicable to clinical data, whose quality is often influenced by the constraints of clinical protocols such as acquisition time, resolution, and hardware limitations. By using clinical standard dMRI data, we aimed to assess the performance of both streamline- and ROI-based segmentation approaches in real-world, clinically relevant scenarios, rather than relying on large-scale open datasets or data acquired under non-clinical conditions. We believe that this approach strengthens the clinical relevance of our study and demonstrates that the streamline-based method can provide reliable and accurate results even under the typical data quality constraints encountered in clinical settings. Accordingly, we expanded the Discussion section of the manuscript (Lines 392-399): “In this work, we chose to perform bundle segmentation in the healthy hemisphere of the patients. This enabled the evaluation of the reliability of the streamline-based approach compared to the ROI-based approach circumventing the debate on the anatomical reliability of bundles deformed by lesions while still testing its performance on data acquired with clinical parameters. By using clinical standard dMRI data, we ensure that our findings are directly applicable to clinical data, whose quality is often influenced by the constraints of clinical protocols such as acquisition time, resolution, and hardware limitations.”We would also like to clarify that segmentations achieved with both ROI-and streamline-based approaches have been carried out on the same whole-brain tractogram for each patient. The whole-brain tractograms have been processed through the same pipeline starting from DWI data acquired with the same parameters for each patient as described in the Methods section. Therefore, the quality of the DWI data has no influence on possible variability in the tracking results within and across patients in this study. 

References

Bertò, G., Bullock, D., Astolfi, P., Hayashi, S., Zigiotto, L., Annicchiarico, L., Corsini, F., De Benedictis, A., Sarubbo, S., Pestilli, F., Avesani, P., & Olivetti, E. (2021). Classifyber, a robust streamline-based linear classifier for white matter bundle segmentation. NeuroImage, 224, 117402. https://doi.org/10.1016/j.neuroimage.2020.117402

Cohen, J. (1969). Statistical power analysis for the behavioral sciences. Academic Press.

Fekonja, L., Wang, Z., Bährend, I., Rosenstock, T., Rösler, J., Wallmeroth, L., Vajkoczy, P., & Picht, T. (2019). Manual for clinical language tractography. Acta Neurochirurgica, 161(6), 1125–1137. https://doi.org/10.1007/s00701-019-03899-0

Forkel, S., Bortolami, C., Dulyan, L., Barrett, R. L., & Beyh, A. (2023). Dissecting white matter pathways: A neuroanatomical approach. https://doi.org/10.31234/osf.io/9xwgm

Ghazi, N., Aarabi, M. H., & Soltanian-Zadeh, H. (2023). Deep Learning Methods for Identification of White Matter Fiber Tracts: Review of State-of-the-Art and Future Prospective. Neuroinformatics, 21(3), 517–548. https://doi.org/10.1007/s12021-023-09636-4

Jeurissen, B., Descoteaux, M., Mori, S., & Leemans, A. (2019). Diffusion MRI fiber tractography of the brain. NMR in Biomedicine, 32(4), e3785. https://doi.org/10.1002/nbm.3785

O’Donnell, L. J., Suter, Y., Rigolo, L., Kahali, P., Zhang, F., Norton, I., Albi, A., Olubiyi, O., Meola, A., Essayed, W. I., Unadkat, P., Ciris, P. A., Wells, W. M., Rathi, Y., Westin, C.-F., & Golby, A. J. (2017). Automated white matter fiber tract identification in patients with brain tumors. NeuroImage: Clinical, 13, 138–153. https://doi.org/10.1016/j.nicl.2016.11.023

Thomas, C., Ye, F. Q., Irfanoglu, M. O., Modi, P., Saleem, K. S., Leopold, D. A., & Pierpaoli, C. (2014). Anatomical accuracy of brain connections derived from diffusion MRI tractography is inherently limited. Proceedings of the National Academy of Sciences, 111(46), 16574–16579. https://doi.org/10.1073/pnas.1405672111

Tournier, J.-D., Calamante, F., & Connelly, A. (2007). Robust determination of the fibre orientation distribution in diffusion MRI: Non-negativity constrained super-resolved spherical deconvolution. NeuroImage, 35(4), 1459–1472. https://doi.org/10.1016/j.neuroimage.2007.02.016

Zigiotto, L., Vavassori, L., Annicchiarico, L., Corsini, F., Avesani, P., Rozzanigo, U., Sarubbo, S., & Papagno, C. (2022). Segregated circuits for phonemic and semantic fluency: A novel patient-tailored disconnection study. NeuroImage: Clinical, 36, 103149. https://doi.org/10.1016/j.nicl.2022.103149

Zoli, M., Talozzi, L., Martinoni, M., Manners, D. N., Badaloni, F., Testa, C., Asioli, S., Mitolo, M., Bartiromo, F., Rochat, M. J., Fabbri, V. P., Sturiale, C., Conti, A., Lodi, R., Mazzatenta, D., & Tonon, C. (2021). From Neurosurgical Planning to Histopathological Brain Tumor Characterization: Potentialities of Arcuate Fasciculus Along-Tract Diffusion Tensor Imaging Tractography Measures. Frontiers in Neurology, 12, 633209. https://doi.org/10.3389/fneur.2021.633209

Reviewer 2 Report

Comments and Suggestions for Authors

Manuscript title

"Changing the paradigm for tractography segmentation in neurosurgery: validation of a streamline-based approach."

1. The main question addressed by the research is whether streamline magnetic resonance tractography is superior to region of interest-based segmentation.

2. The manuscript may be considered relevant to neuroimaging. However, the importance of this original study is limited by small sample size, statistical considerations as well as clinical endpoint.

3. The manuscript expands prior published material by demonstrating that streamline-based segmentation may yield lower irregularity scores.

4. The specific improvements to methodology include:

  • Clarifying the relationship between preoperative WM anatomy visualization and patient outcomes (levels of evidence? clinical guidelines? consider https://eurradiolexp.springeropen.com/articles/10.1186/s41747-018-0066-1);

  • Supplementing the information on current diffusion tensor imaging quality metrics and rationale for selecting only irregularity score (consider https://pmc.ncbi.nlm.nih.gov/articles/PMC4753778/);

  • Including possible conflict of interest or revising the manuscript to include that Paolo Avesani was one of the Tractome developers (https://github.com/FBK-NILab/tractome);

  • Providing rationale for using only a healthy hemisphere in the study, as possible Tractome benefits should be demonstrated near the pathological lesion;

  • Providing sample size calculation, as 25 patient may be considered too few to obtain reproducible data; why were open access datasets not used, such as https://openneuro.org/datasets/ds001378/versions/00003 and https://www.nature.com/articles/s41597-021-01092-6;

  • Providing information on data normality testing (using Shapiro-Wilk test) to select appropriate test (parametric or non-parametric, possibly with Benjamini-Hochberg procedure);

  • Providing information on the set level of statistical significance for p-value as well as statistical package used.

  • Expanding the Discussion section with study limitations.

5. The conclusions are not fully consistent with the evidence provided, given methodological and statistical considerations.

6. The references are appropriate.

7. The figures and tables are appropriate.

Comments on the Quality of English Language

Consider performing and additional check using LanguageTool or similar extension, as well as putting the references in brackets.

Author Response

Comment 1: The main question addressed by the research is whether streamline magnetic resonance tractography is superior to region of interest-based segmentation.

Response 1: We thank Reviewer 2 for taking the time to review this manuscript and for highlighting the central question of this study. The aim of the present study is indeed to compare the performance of two approaches for bundle segmentation, namely ROI- and streamline-based approaches. Our analyses were carried out on data acquired according to clinical parameters, not solely to determine which method is superior but to ensure that our findings are directly applicable to clinical data, whose quality is often influenced by the constraints of clinical protocols. 

Comment 2: The manuscript may be considered relevant to neuroimaging. However, the importance of this original study is limited by small sample size, statistical considerations as well as clinical endpoint.

Response 2: We are grateful to Reviewer 2 for his comments and for recognizing the relevance of our study to the neuroimaging community. As you will find in the detailed responses below and in the revisions highlighted in red in the re-submitted file, we incorporated your useful suggestion to improve our manuscript, tackling both the statistical considerations and the clinical endpoint of this work. 

Comment 3: The manuscript expands prior published material by demonstrating that streamline-based segmentation may yield lower irregularity scores.

Response 3: We thank Reviewer 2 for this observation and for recognizing the contribution of our manuscript in expanding prior published material. While the previous work by Porro-Muñoz et al., 2015 illustrates the software Tractome, the present work investigates its impact and added value compared to best-practice approaches, such as the ROI-based approach on standard clinical data. 

Comment 4.1:The specific improvements to methodology include:

Clarifying the relationship between preoperative WM anatomy visualization and patient outcomes (levels of evidence? clinical guidelines? consider https://eurradiolexp.springeropen.com/articles/10.1186/s41747-018-0066-1);

Response 4.1: We thank Reviewer 2 for highlighting the value of establishing a relationship between preoperative images, or WM bundle segmentations in the case of our work, and the clinical outcome of the patient. While we recognize the critical role that accurate WM bundle segmentations play in improving presurgical planning, intraoperative navigation and therefore clinical outcomes, we would like to clarify that the primary goal of our study was not to directly assess the clinical impact of the results of WM bundle segmentation on the patients’ outcome. Instead, our focus was on comparing two methodological approaches (i.e., streamline- and ROI-based) for bundle segmentation in clinical-like settings, using data acquired with a standard clinical imaging protocol. By demonstrating the methodological strengths of the streamline-based approach, we want to provide clinicians with the normative validation of a tool, carried out on data whose quality is generally influenced by clinical constraints, that could enhance their ability to interpret and use tractography for bundle segmentation in challenging scenarios. Regarding the broader implications of the study, we agree that improved and more anatomically reliable representation of WM bundles has the potential to benefit, or at least better predict, patients’ outcome. Indeed, evidence linking preoperative WM bundle reconstructions in tractography to surgical outcome is growing (e.g., Zigiotto et al., 2022; Coletta et al., 2024). However, establishing such a relationship was beyond the scope of the present study. Instead our findings contribute to refining the methodological basis upon which future investigations and clinical applications could build. We further clarify this point and indicate it as a future perspective in the Discussion section as follows (Lines 418-424): “As we demonstrate the methodological strengths of a streamline-based approach for bundle segmentation on standard clinical data, we set the basis for future investigations and clinical applications. Future studies are needed to apply our comparative approach to lesioned hemispheres, and to investigate whether the use of bundle segmentations obtained with streamline-based approaches carries clinical benefits, such as a better prediction of the clinical outcome of patients, compared to ROI-based segmentations.”  

Comment 4.2: Supplementing the information on current diffusion tensor imaging quality metrics and rationale for selecting only irregularity score (consider https://pmc.ncbi.nlm.nih.gov/articles/PMC4753778/);

Response 4.2: We appreciate Reviewer 2’s suggestion to consider additional quality metrics. We would like to clarify that there is a distinction between the quality assurance of DWI images, or even of DTI images as pointed out in the paper referenced by the Reviewer, and the evaluation of bundle segmentations. DWI/DTI quality assurance pertains to the data collection and the computational pre-processing stages, where factors like signal-to-noise ratio, spatial resolution and motion artefacts are evaluated. We acknowledge these aspects to be fundamental for ensuring accuracy of the diffusion data and the tractography model underlying the computation of streamlines (Thomas et al., 2014). However, our study focuses on the evaluation of the results of manual bundle segmentation, which occurs after the reconstruction of streamlines from DWI images, signal modelling, and the assumption of a tracking algorithm. Moreover, our study does not employ DTI-based tractography, but rather CSD tractography. CSD is considered a more advanced modelling method that better handles crossing fiber populations and provides more reliable tractography results compared to DTI. 

Regarding the irregularity index, this measurement was introduced by Yeh, 2020, establishing a specific framework for the evaluation of WM bundles segmented from tractography. Among several metrics proposed (e.g., length, span, diameter, radius), the irregularity index better represents the overall shape of a bundle, factoring in its surface area, diameter, and length. In particular, we found this index to be the best fit currently available in the literature to quantify the presence of streamlines deviating from the main overall course of the bundle, since it exemplifies its compactness. Hence the use of this index to compare the results of streamline- and ROI-based segmentations achieved in the same patient starting from the same whole-brain tractogram. Based on Reviewer 2’s suggestion, we now expanded on the choice of the irregularity index for the comparison of the results of the two bundle segmentation approaches in the Methods section of the manuscript (Lines 257-262): “Among the several metrics proposed in this framework, the irregularity measurement defines the compactness of a bundle, factoring in its surface area, diameter and length. This index is sensitive to the presence of streamlines deviating from the overall main course of the bundle, with lower irregularity values indicating more compact bundles. We therefore selected the irregularity measurement to compare the segmentation of the same bundle in the same patient achieved through ROI- and streamline- based approaches.”.

Comment 4.3: Including possible conflict of interest or revising the manuscript to include that Paolo Avesani was one of the Tractome developers (https://github.com/FBK-NILab/tractome);

Response 4.3: We thank Reviewer 2 for pointing out the potential conflict of interest regarding Paolo Avesani’s involvement in the development of the software Tractome. To address this concern, we have clarified the author’s role and position in the Conflict of Interest section of the manuscript. The revised statement reads (Lines 466-469): “P.A. contributed to the development of the software Tractome used in this study. However, he has no financial or other competing interests related to its use or the outcomes of this research. Additionally, he did not participate in the process of bundle segmentation and was blind to the segmentation results.”

Comment 4.4: Providing rationale for using only a healthy hemisphere in the study, as possible Tractome benefits should be demonstrated near the pathological lesion;

Response 4.4: We thank Reviewer 2 for highlighting this crucial point. As we stated in the Discussion section (Lines 392-399 of the current version of the manuscript, adapted accordingly to the Reviewers’ comments), “In this work, we chose to perform bundle segmentation in the healthy hemisphere of the patients. This enabled the evaluation of the reliability of the streamline-based approach compared to the ROI-based approach circumventing the debate on the anatomical reliability of bundles deformed by lesions while still testing its performance on data acquired with clinical parameters. By using clinical standard dMRI data, we ensure that our findings are directly applicable to clinical data, whose quality is often influenced by the constraints of clinical protocols such as acquisition time, resolution, and hardware limitations.” Indeed, the objective of the present study is to evaluate the performance of the two different approaches for bundle segmentation on data acquired following clinical standards while setting aside controversies related to the assessment of what is the real anatomy in the presence of morphological brain alterations. Indeed, to date, there is no precise criterion defining plausible and implausible streamlines nearby the lesion site, and streamlines that would normally be defined as anatomically implausible due to major deviations from the expected course of a bundle might be reasonable in the presence of a lesion. We agree that this step forward should be taken by future studies to demonstrate that the features that in principle make Tractome an optimal tool for bundle segmentation in the clinical setting provide the same benefits that we described in the present work in non-distorted anatomies. Thanks to Reviewer 2’s comment, we now further extended this point in the Limitations section (Lines 427-439) : “Additionally, this study was conducted on the healthy hemisphere of the patients we selected. This choice was aimed to test and describe the differences in bundle segmentations that can be achieved via ROI- and streamline-based approaches in optimal settings, setting aside the debate related to what is the ground truth of anatomical distortions in the presence of a lesion, while still using data acquired with clinical standards. Future works including hemispheres affected by lesions are needed to better demonstrate the potential benefits of Tractome for bundle segmentation in distorted anatomies.” 

Comment 4.5: Providing sample size calculation, as 25 patient may be considered too few to obtain reproducible data; why were open access datasets not used, such as https://openneuro.org/datasets/ds001378/versions/00003 and https://www.nature.com/articles/s41597-021-01092-6;

Providing information on data normality testing (using Shapiro-Wilk test) to select appropriate test (parametric or non-parametric, possibly with Benjamini-Hochberg procedure);

Providing information on the set level of statistical significance for p-value as well as statistical package used.

Response 4.5: We appreciate the observation of Reviewer 2 regarding the sample size and the statistics of our study. While we acknowledge that the sample size is relatively small, it is consistent with other clinical studies performing tractography bundle segmentation on tumor patients (e.g., O’Donnel et al., 2017, Zoli et al., 2021, Fekonja et al., 2019). We thank Reviewer 2 for pointing out two interesting open source datasets on which the same approach we used in the present work could be applied. Nevertheless, we stress that, as reported in the Discussion section of the manuscript (Lines 392-399), this study aimed to compare the performance of the two segmentation approaches (i.e., ROI- and streamline-based) on data acquired with clinical standards: “we chose to perform bundle segmentation in the healthy hemisphere of the patients. This enabled the evaluation of the reliability of the streamline-based approach compared to the ROI-based approach circumventing the debate on the anatomical reliability of bundles deformed by lesions while still testing its performance on data acquired with clinical parameters. By using clinical standard dMRI data, we ensure that our findings are directly applicable to clinical data, whose quality is often influenced by the constraints of clinical protocols such as acquisition time, resolution, and hardware limitations.”. Following this comment, we report the calculation of Cohen’s d (Cohen, 1969) to quantify the effect size of the paired t-test in Table 2, in the Methods (Line 266) and in the Results section (Line 293). To address concerns about generalizability of findings, we have included the following acknowledgement in the Limitations section (Lines 427-432): “We acknowledge that the sample size of this study is relatively small, which may limit the generalizability of our findings. Despite this limitation, the significant differences observed between the ROI- and streamline-based approaches, supported by the calculation of Cohen’s d for effect size, provide robust evidence for the reliability of our findings. Future studies with larger and more diverse cohorts will be critical to further validate and extend the applicability of our conclusions to other clinical and research contexts.” 

Thanks to Reviewer 2’s comment, we now describe the normality of the distribution of our data based on the results of the Shapiro-Wilk test (p>0.05), justifying the choice of selecting a parametric test (Shapiro & Wilk, 1965). We therefore added in the Methods section (Lines 262-264): “Given the normal distribution of the data (p>0.05 for Shapiro-Wilk test), parametric testing was performed.”. Furthermore, we provide explicit information on the threshold used for statistical significance of p-values and integrated information about the program used for the analyses (see Lines 266-268): “Analyses were performed in Python 3.10, and the significance level for paired t-test was set at p<0.05.”.

Comment 4.6: Expanding the Discussion section with study limitations.

Response 4.6: Thanks to this comment, a Limitation section has been added to the manuscript (Lines 426-439). “We acknowledge that the sample size of this study is relatively small, which may limit the generalizability of our findings. Despite this limitation, the significant differences observed between the ROI- and streamline-based approaches, supported by the calculation of Cohen’s d for effect size, provide robust evidence for the reliability of our findings. Future studies with larger and more diverse cohorts will be critical to further validate and extend the applicability of our conclusions to other clinical and research contexts. Additionally, this study was conducted on the healthy hemisphere of the patients we selected. This choice was aimed to test and describe the differences in bundle segmentations that can be achieved via ROI- and streamline-based approaches in optimal settings, setting aside the debate related to what is the ground truth of anatomical distortions in the presence of a lesion, while still using data acquired with clinical standards. Future works including hemispheres affected by lesions are needed to better demonstrate the potential benefits of streamline-based approaches for bundle segmentation in distorted anatomies.”

Comment 5: The conclusions are not fully consistent with the evidence provided, given methodological and statistical considerations.

Response 5: We thank Reviewer 2 for pointing out that the conclusions should align closely with the evidence provided, considering methodological and statistical aspects. While we stand by the validity of our findings, we acknowledge the need of ensuring that our conclusions are explicitly grounded in the results presented. Our study demonstrates the methodological advantages of the streamline-based approach for bundle segmentation, as implemented in Tractome, compared to the ROI-based approach. Specifically, both the quantitative and qualitative analyses support the claim that the streamline-based approach provides more compact and anatomically accurate reconstructions of WM bundles. To address Reviewer 2’s comment, we have revised the Conclusion section to better reflect the evidence provided. The updated version of the Conclusion section reads as follows (Lines 442-449): “The present study demonstrates that streamline-based bundle segmentation, as implemented in the software Tractome, provides more compact and anatomically reliable representations of WM bundles compared to the traditional ROI-based approach, based on quantitative irregularity scores and qualitative assessment of streamlines plausibility. These findings highlight methodological advantages that may support clinicians in visualizing and interpreting WM anatomy, particularly in the presence of lesions that deform the anatomy of the brain. Future studies will be required to validate these findings across broader datasets and assess their impact on different clinical scenarios.”

Comment 6: The references are appropriate.

Response 6: We thank Reviewer 2 for the positive feedback regarding the references.

Comment 7: The figures and tables are appropriate.

Response 7: We thank Reviewer 2 for the positive feedback regarding the figures and tables.

As suggested by Reviewer 2, we performed language check with the software LanguageTool and implemented corrections in the manuscript.

References

Cohen, J. (1969). Statistical power analysis for the behavioral sciences. Academic Press.

Coletta, L., Avesani, P., Zigiotto, L., Venturini, M., Annicchiarico, L., Vavassori, L., Ng, S., Duffau, H., & Sarubbo, S. (2024). Integrating direct electrical brain stimulation with the human connectome. Brain, 147(3), 1100–1111. https://doi.org/10.1093/brain/awad402

Fekonja, L., Wang, Z., Bährend, I., Rosenstock, T., Rösler, J., Wallmeroth, L., Vajkoczy, P., & Picht, T. (2019). Manual for clinical language tractography. Acta Neurochirurgica, 161(6), 1125–1137. https://doi.org/10.1007/s00701-019-03899-0

O’Donnell, L. J., Suter, Y., Rigolo, L., Kahali, P., Zhang, F., Norton, I., Albi, A., Olubiyi, O., Meola, A., Essayed, W. I., Unadkat, P., Ciris, P. A., Wells, W. M., Rathi, Y., Westin, C.-F., & Golby, A. J. (2017). Automated white matter fiber tract identification in patients with brain tumors. NeuroImage: Clinical, 13, 138–153. https://doi.org/10.1016/j.nicl.2016.11.023

Porro-Muñoz, E. Olivetti, N. Sharmin, T. B. Nguyen, E. Garyfallidis, and P. Avesani, ‘Tractome: a visual data mining tool for brain connectivity analysis’, Data Mining and Knowledge Discovery, vol. 29, pp. 1258–1279, 2015.

Shapiro, S. S., & Wilk, M. B. (1965). An analysis of variance test for normality. Biometrika, 52(3-4), 591–611. https://doi.org/10.1093/biomet/52.3-4.591

Thomas, C., Ye, F. Q., Irfanoglu, M. O., Modi, P., Saleem, K. S., Leopold, D. A., & Pierpaoli, C. (2014). Anatomical accuracy of brain connections derived from diffusion MRI tractography is inherently limited. Proceedings of the National Academy of Sciences, 111(46), 16574–16579. https://doi.org/10.1073/pnas.1405672111

Yeh, F.-C. (2020). Shape analysis of the human association pathways. NeuroImage, 223, 117329. https://doi.org/10.1016/j.neuroimage.2020.117329

Zigiotto, L., Vavassori, L., Annicchiarico, L., Corsini, F., Avesani, P., Rozzanigo, U., Sarubbo, S., & Papagno, C. (2022). Segregated circuits for phonemic and semantic fluency: A novel patient-tailored disconnection study. NeuroImage: Clinical, 36, 103149. https://doi.org/10.1016/j.nicl.2022.103149

Zoli, M., Talozzi, L., Martinoni, M., Manners, D. N., Badaloni, F., Testa, C., Asioli, S., Mitolo, M., Bartiromo, F., Rochat, M. J., Fabbri, V. P., Sturiale, C., Conti, A., Lodi, R., Mazzatenta, D., & Tonon, C. (2021). From Neurosurgical Planning to Histopathological Brain Tumor Characterization: Potentialities of Arcuate Fasciculus Along-Tract Diffusion Tensor Imaging Tractography Measures. Frontiers in Neurology, 12, 633209. https://doi.org/10.3389/fneur.2021.633209

Round 2

Reviewer 1 Report

Comments and Suggestions for Authors

The author has provided a satisfactory response to the question and I recommend publication.

Reviewer 2 Report

Comments and Suggestions for Authors

The authors have provided point-by-point responses, fully addressing the reviewer's queries and further improving the manuscript's quality.